physical chemistry

aggregated system, amino acid, gemini surfactants, interfaces, micelles, ninhydrin

**Author for correspondence:**
Dileep Kumar
e-mail: dileepkumar@tdtu.edu.vn

This article has been edited by the Royal Society of Chemistry, including the commissioning, peer review process and editorial aspects up to the point of acceptance.

# Catalytic influence of 16-*s*-16 gemini surfactants on the rate constant of histidine and ninhydrin

Dileep Kumar[1,2] and Malik Abdul Rub[3]

[1]Division of Computational Physics, Institute for Computational Science, Ton Duc Thang University, Ho Chi Minh City, Vietnam
[2]Faculty of Applied Sciences, Ton Duc Thang University, Ho Chi Minh City, Vietnam
[3]Chemistry Department, Faculty of Science, King Abdulaziz University, Jeddah 21589, Saudi Arabia

 DK, 0000-0003-2913-5032

The present paper reports the catalytic influence of 16-*s*-16 (spacer (*s*) = 4, 5, 6) gemini surfactants on the rate constant of histidine and ninhydrin at 343 K and pH 5.0 using the spectrophotometric technique. The effect of varying amounts of geminis was made on the rate constant of histidine and ninhydrin keeping other constituents constant. Characteristics of the rate constant ($k_\psi$) versus [gemini] depict the effect of surfactants on the rate constant. A systematic explanation about the effect of surfactants is revealed and discussed in the text. The influence of different parameters that includes [reactants], temperature and pH has also been performed on the study. In order to determine the critical micelle concentration (cmc) of pure surfactants and their solution mixtures, conductivity measurement was employed. By using the Eyring equation, activation parameters at different temperatures have been obtained. The resultant data of $k_\psi$ versus [gemini] plot were rationalized with the pseudo-phase model of micelles.

## 1. Introduction

Surfactants contain a polar head (hydrophilic group) and a non-polar (hydrophobic hydrocarbon tail). They have been employed in several aspects of chemical, biological and industrial processes [1–5]. Surfactants aggregate frequently and catalyse the chemical reactions, but they also inhibit reactions. Micelles affect reaction rate owing to numerous aspects [6]. A little alteration in the structure of micelle induces variations in surface properties and rigidity of micelles; consequently, it alters the reactants' activity [7–9]. Micelles offer several unusual media for different parts of the reactants. Solubilization of micellar environment could serve as a key part in micellar catalysis of a chemical

reaction. The rate of bimolecular reactions increases in ionic micelles by increasing the concentration of the reactant molecules into the Stern layer (a small volume). By considering the electrostatic and hydrophobic interactions between reactant molecules and micelles, the kinetic effect on the reactions can be accounted in micellar media.

Gemini surfactant comprises two traditional monomeric surfactants connected by a spacer [10–15]. Compared with conventional surfactants (single head and single chain), gemini surfactants show unique physico-chemical behaviour that depends upon the nature of hydrophilic head, hydrophobic tail and spacer chain length. They frequently exhibit the special physical properties, such as lower critical micelle concentration (cmc) value, enhanced surface active power and rich aggregation properties compared with those of the corresponding conventional surfactants [16–22]. Owing to their outstanding aggregation character and surface properties, they have been used in very high potential applications in daily household chemicals, coatings, petroleum and so on [23,24]. Geminis have much lower cmc value in comparison with conventional ones. Being low cmc value, gemini surfactant implies as a high cost-effective surfactant. From an environmental point of view, they can be regarded as a green surfactant due to the use of smaller surfactant quantities, which are in line with the green surfactant idea in surfactant chemistry [25,26]. The nature, mode of action and mechanism of 16-$s$-16 surfactant on the interaction of ninhydrin with histidine may be a matter of great consideration.

2,2-Dihydroxy-1,3-indandione, commonly known as ninhydrin, is a universal colour developing agent. It has been widely used due to its distinctive sensitivity for the identification of amino group in chemistry, biochemistry and bio-analytical study. Ninhydrin interacts with the amino group and forms a purple-coloured product diketohydrindylidenediketohydrindamine (DYDA) [27,28]; ([29] and references therein). As DYDA colour fades, numerous modifications that include the effect of various traditional surfactants, different solvents and pretreatment of enzymes to stabilize the product were performed [30–35].

Although a number of modifications on the reaction of ninhydrin with amino acid have been brought to increase the stability and potential applicability of DYDA product, but the problem related to studies with 16-$s$-16 gemini surfactants under different situations remains unexplored.

Therefore, to get further response and better output, we have reported the catalytic influence of 16-$s$-16 on the rate constant of histidine and ninhydrin reaction. Studies of different constituents including the effect of reactants, temperature and pH were also made in 16-$s$-16 surfactants.

# 2. Experimental procedure

## 2.1. Material and method

Freshly prepared double-distilled water (conductivity: 1–2 $\mu$S cm$^{-1}$) was used in all physico-chemical measurements throughout the experiments. All the reagents used for synthesizing the gemini surfactants were N,N-dimethylcetylamine (greater than 95.0%), 1,4-dibromobutane (greater than 98%), 1,5-dibromopentane (greater than 98%) and 1,6-dibromohexane (greater than 97%). They were used as obtained (Fluka, Germany). L-Histidine (99.0%) was used as received from Loba Chemie, India. Ninhydrin (99.0%), CH$_3$COOH (99.0%) and CH$_3$COONa (99.0%) were purchased from Merck and applied without any further purification. The rest of the chemicals used were of the best AR grade. The buffer solution was used for preparation of all standard solutions of ninhydrin, histidine and gemini surfactants [36]. All the pH analyses were made with an ELICO digital pH meter.

Surfactants were synthesized via method described earlier [37]. N,N-dimethylcetylamine and $\alpha,\omega$-dibromoalkane were placed in a double-necked round-bottomed 2 l flask. The mixture was stirred regularly using a magnetic bar at 353 K for 2 days to get complete bisquaternization. After the reaction was accomplished, solvent was removed. Thus, solid obtained was purified three times by the mixed solvent of hexane and ethyl acetate. Then, the crude product was dried under vacuum to get a white solid as a final product (gemini surfactants). The identification of the final product was checked by proton magnetic resonance spectra ($^1$H NMR) and elemental analyses. Data obtained are consistent with those described previously [37].

## 2.2. Determination of electrical conductivity

The specific conductivity of pure gemini surfactants and their solution mixtures as a function of [surfactant] was recorded using Systronics conductivity meter (Ahmedabad, India). [Surfactant] was added gradually

to the glass flask containing pure water/mixture. To provide a homogeneous environment to the system, the solution mixture was stirred continuously after each addition. Specific conductivities were noted down at 303 and 343 K. All analyses at each temperature were made at a minimum of three runs. The cmc value was estimated from the intersection of two linear intercepts between plots of specific conductance versus [surfactant] [38–40]. The determined cmc values of geminis with and without reactants (i.e. water, water + ninhydrin, water + ninhydrin + histidine) are provided below.

(a) [16-6-16]: $0.043 \times 10^{-3}$, $0.039 \times 10^{-3}$ and $0.038 \times 10^{-3}$ mol dm$^{-3}$ at 303 K; $0.058 \times 10^{-3}$, $0.055 \times 10^{-3}$ and $0.052 \times 10^{-3}$ mol dm$^{-3}$ at 343 K.
(b) [16-5-16]: $0.034 \times 10^{-3}$, $0.033 \times 10^{-3}$ and $0.032 \times 10^{-3}$ mol dm$^{-3}$ at 303 K; $0.050 \times 10^{-3}$, $0.043 \times 10^{-3}$ and $0.048 \times 10^{-3}$ mol dm$^{-3}$ at 343 K.
(c) [16-4-16]: $0.032 \times 10^{-3}$, $0.031 \times 10^{-3}$ and $0.028 \times 10^{-3}$ mol dm$^{-3}$ at 303 K; $0.044 \times 10^{-3}$, $0.040 \times 10^{-3}$ and $0.038 \times 10^{-3}$ mol dm$^{-3}$ at 343 K.

## 2.3. Kinetic measurements

Measurements were made under pseudo-first-order condition where [ninhydrin] was kept 60 times more to [His]. Suitable amount of histidine, buffers and gemini surfactant (when required) was placed in a thermostat-controlled water bath. Mixture was kept at an experimental temperature and left for 30 min for equilibration. To start the reaction quickly, a known volume of thermally equilibrated ninhydrin solution was poured into the mixture which was placed separately in the same thermostat-controlled bath. Absorbance was noted at a regular time interval at maximum wavelength using SHIMADZU model spectrophotometer (Kyoto, Japan). Rate constants were calculated by employing a computer-based procedure. Other information related to kinetic measurements can be found in the existing literature reported previously [41–46].

# 3. Results

## 3.1. Spectra

Spectra on the rate constant of histidine and ninhydrin in pure water and surfactants were measured after completion of reaction over a wavelength range 350–650 nm using a quartz cuvette with 1 cm path length (figure 1; electronic supplementary material, table S1). It was noted from experimental results that the values of absorbance were increased on increasing [gemini] without any change in maximum wavelength ($\lambda_{max} = 570$ nm); inferring the same product formed in two systems, i.e. remains unchanged.

## 3.2. Effect of several parameters on $k_\psi$

The influence of pH on $k_\psi$ on ninhydrin and histidine reaction was seen at several pH varying from 4.0 to 6.0 at constant histidine, ninhydrin and temperature (343 K) in gemini system ([16-$s$-16] = $30 \times 10^{-5}$ mol dm$^{-3}$) (table 1). $k_\psi$ increases with pH up to 5.0. Beyond pH 5.0, the rate becomes almost unchanging, i.e. remains almost constant. $k_\psi$ values corresponding to different pH are plotted and shown in electronic supplementary material, figure S1. This infers that Schiff base is formed in the vicinity of pH (5.0), since Schiff base is an acid-catalysed reaction and optimum pH is 5.0. So, all the analyses were undertaken at pH 5.0.

Experiments were made at different initial [His] in gemini micellar media under the set of identical reaction situation of ninhydrin, pH and temperature. The resultant values of $k_\psi$ at different [His] are given in table 1 and shown graphically in electronic supplementary material, figure S2. The rate constant ($k_\psi$) suggests that $k_\psi$ does not depend on initial [His]; establishing the order to be unity in [His].

Rate constants were determined at various [ninhydrin] on the present reaction keeping other reaction ingredients constant (electronic supplementary material, table S2). The rate profile of $k_\psi$ values against [ninhydrin] is depicted in figure 2. It can be seen that the plot of figure 2 provides a nonlinear profile and crosses through the origin. The above leads fractional-order in relation to ninhydrin.

$k_\psi$ values were achieved in gemini surfactants by performing the studies at several temperatures (333–353 K) with an interval of 5 K at fixed [reactants] (ninhydrin and histidine) and pH. The observed $k_\psi$ values are tabulated (table 1). By using these results, different activation parameters were calculated by linear least-squares regression procedure.

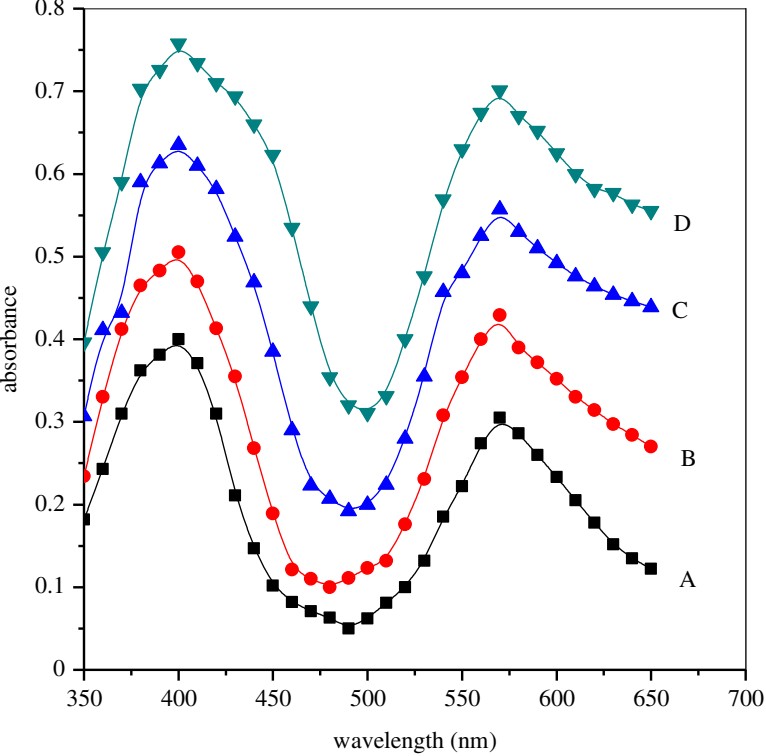

**Figure 1.** Spectra of product formed on the study of histidine and ninhydrin in pure water and surfactants at 343 K and pH = 5.0 after heating 2 h: (A) aqueous, (B) 16-6-16, (C) 16-5-16 and (D) 16-4-16. Reaction conditions: [His] = $1 \times 10^{-4}$ mol dm$^{-3}$, [ninhydrin] = $6.0 \times 10^{-3}$ mol dm$^{-3}$ and [16-$s$-16] = $30 \times 10^{-5}$ mol dm$^{-3}$.

**Table 1.** Influence of different parameters on $k_\psi$ on the study of histidine and ninhydrin at [Nin] ($6 \times 10^{-3}$ mol dm$^{-3}$) in 16-$s$-16 gemini surfactants ($30 \times 10^{-5}$ mol dm$^{-3}$). Uncertainties in $k_\psi$ values are estimated to be less than or equal to $\pm 0.1 \times 10^{-4}$ s$^{-1}$.

| $10^4$ [His] (mol dm$^{-3}$) | pH | Temp. (K) | $10^4 \, k_\psi$ (s$^{-1}$) | | |
|---|---|---|---|---|---|
| | | | 16-6-16 | 16-5-16 | 16-4-16 |
| 1.0 | 5.0 | 343 | 5.5 | 6.5 | 7.7 |
| 1.5 | | | 5.5 | 6.5 | 7.6 |
| 2.0 | | | 5.4 | 6.5 | 7.7 |
| 2.5 | | | 5.4 | 6.4 | 7.6 |
| 3.0 | | | 5.5 | 6.4 | 7.7 |
| 1.0 | 4.0 | 343 | 2.2 | 3.1 | 3.8 |
| | 4.5 | | 3.4 | 4.3 | 5.6 |
| | 5.0 | | 5.5 | 6.5 | 7.7 |
| | 5.5 | | 6.0 | 6.9 | 8.1 |
| | 6.0 | | 6.2 | 7.1 | 8.4 |
| 1.0 | 5.0 | 333 | 4.0 | 4.8 | 6.3 |
| | | 338 | 4.6 | 5.5 | 6.9 |
| | | 343 | 5.5 | 6.5 | 7.7 |
| | | 348 | 6.7 | 7.9 | 9.3 |
| | | 353 | 8.2 | 9.7 | 10.9 |

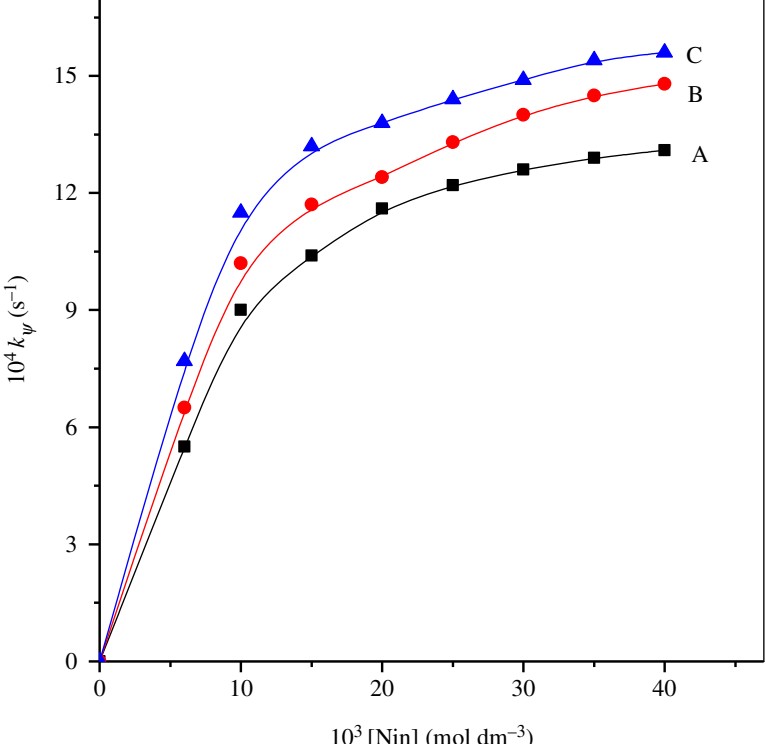

**Figure 2.** Rate constant ($k_\psi$) versus [Nin] on the study of histidine and ninhydrin in surfactants: (A) 16-6-16, (B) 16-5-16 and (C) 16-4-16. Reaction conditions: [His] $= 1 \times 10^{-4}$ mol dm$^{-3}$, [16-$s$-16] $= 30 \times 10^{-5}$ mol dm$^{-3}$, temperature $= 343$ K and pH $= 5.0$.

## 4. Discussion

### 4.1. Proposed reaction mechanism between ninhydrin and histidine

The mechanism of amino acids with ninhydrin has received great attention of researchers/investigators due to the use in different fields [47–51]. The proposed mechanism for the current study of ninhydrin and histidine is demonstrated in scheme 1. The reaction has been observed to start through the formation of DYDA involving the mechanism of reaction that follows two steps. (i) Condensation between carbonyl group (of ninhydrin) and amino group occurs, which produces an intermediate Schiff base (after decarboxylation). This intermediate is highly unstable and hydrolyses to result in 2-aminoindanedione and an aldehyde. (ii) The reaction of 2-aminoindanedione connects slowly with ninhydrin to produce DYDA.

### 4.2. Reaction in 16-$s$-16 gemini micellar media

Rate constants were obtained by varying [16-$s$-16] in the range $(5–3000) \times 10^{-5}$ mol dm$^{-3}$ keeping other reaction factors constant. These are given in table 2. The outcomes suggested that the same first- and fractional-order paths were detected in [His] and [ninhydrin], respectively, in the presence of surfactants as that of pure water. The above observation of the same product formation confirms that the mechanism remains exactly identical in both the systems. Variation of $k_\psi$ with [gemini] is plotted and presented graphically in figure 3.

The catalytic influence of gemini on the $k_\psi$ on histidine and ninhydrin reaction can be interpret by the means of pseudo-phase model (scheme 2), suggested by Menger & Portnoy [52] and established by Bunton [53] and Romsted [54].

$r = k_\psi$ ([His]) and scheme 2 result in the below equation

$$k_\psi = \frac{k_w' + k_m' K_C[\text{gemini} - \text{cmc}]}{1 + K_C[\text{gemini} - \text{cmc}]}. \tag{4.1}$$

Then, the above equation converted into the below equation

$$k_\psi = \frac{k_w[\text{Nin}]_T + (K_C k_m - k_w) M_N^S[\text{gemini} - \text{cmc}]}{1 + K_C[\text{gemini} - \text{cmc}]}. \tag{4.2}$$

**Scheme 1.** Mechanism for the study of ninhydrin and histidine reaction. $K$ and $k$ stand for the equilibrium constant and rate constant, respectively.

Herein, $K_C$ and $K_D$ designate the respective binding constant for histidine and ninhydrin. $M_N^S = [\text{Nin}]_m/[\text{gemini} - \text{cmc}]$ is [ninhydrin] in the molar ratio of the micellar head group. $k_w(= k_w'/[\text{Nin}]_w)$ and $k_m(= k_m'/M_N^S)$ are symbolized as second-order rate constants, respectively. The determination of $K_C$ and $k_m$ needs cmc data under experimental kinetic conditions. Therefore, cmc was calculated using the conductivity meter. For a given cmc, $K_C$ and $k_m$ were achieved from equation (4.2) by the computer procedure. The calculated values of the rate constant ($k_{\psi\text{cal}}$) were obtained by substituting $K_C$ and $k_m$ in rate equation (4.2). The clear matching between the values of $k_\psi$ and $k_{\psi\text{cal}}$ within instrumental errors validates the proposed reaction mechanism (table 2) (electronic supplementary material, table S3).

Considering segment I (figure 3), where [16-*s*-16] are much lower than their cmc value, rate values should not enhance. The enhancement in rate may take place owing to the formation of premicelle aggregates between surfactants and reactants [55,56]. It is well established that gemini surfactant can produce various aggregates with different additives, viz., micelles, bilayers, vesicles, etc. The formation of premicelle aggregates between reactant and surfactant molecules and their catalytic behaviour can be found in the published literature [57,58].

Segment II, figure 3, [16-*s*-16] is up to $400 \times 10^{-5}$ mol dm$^{-3}$. No variation in $k_\psi$ values was detected, i.e. almost invariant and followed the order of their catalysing effect: 16-4-16 > 16-5-16 > 16-6-16. Gemini micellar systems lead to significantly better catalysing properties when compared with conventional

**Table 2.** Influence of [gemini] on $k_\psi$ on the study of histidine $(1.0 \times 10^{-4}$ mol dm$^{-3})$ and ninhydrin $(6.0 \times 10^{-3}$ mol dm$^{-3})$ at temperature (343 K) and pH (5.0); and their comparison with $k_{\psi cal}$. Uncertainties in the values of the rate constant are estimated to be less than or equal to $\pm 0.1 \times 10^{-4}$ s$^{-1}$.

| $10^5$ [gemini] (mol dm$^{-3}$) | 16-6-16 | | | 16-5-16 | | | 16-4-16 | | |
|---|---|---|---|---|---|---|---|---|---|
| | $10^4\,k_\psi$ (s$^{-1}$) | $10^4\,k_{\psi cal}$ (s$^{-1}$) | $\frac{k_\psi - k_{\psi cal}}{k_\psi}$ | $10^4\,k_\psi$ (s$^{-1}$) | $10^4\,k_{\psi cal}$ (s$^{-1}$) | $\frac{k_\psi - k_{\psi cal}}{k_\psi}$ | $10^4\,k_\psi$ (s$^{-1}$) | $10^4\,k_{\psi cal}$ (s$^{-1}$) | $\frac{k_\psi - k_{\psi cal}}{k_\psi}$ |
| 0.0 | 1.6 | — | — | 1.6 | — | — | 1.6 | — | — |
| 5.0 | 2.6 | — | — | 3.7 | — | — | 4.3 | — | — |
| 10.0 | 3.4 | 3.5 | −0.029 | 4.2 | 4.3 | −0.023 | 5.4 | 5.2 | +0.037 |
| 20.0 | 4.7 | 4.6 | +0.021 | 5.4 | 5.6 | −0.037 | 6.8 | 6.5 | +0.044 |
| 30.0 | 5.5 | 5.5 | 0 | 6.5 | 6.4 | +0.015 | 7.7 | 8.0 | −0.038 |
| 40.0 | 5.7 | 5.9 | −0.035 | 6.7 | 6.5 | +0.029 | 7.9 | 7.9 | 0 |
| 50.0 | 5.8 | 6.0 | −0.034 | 6.8 | 6.6 | +0.029 | 8.0 | 8.0 | 0 |
| 60.0 | 6.0 | 5.8 | +0.033 | 6.9 | 7.1 | −0.028 | 8.2 | 8.4 | −0.024 |
| 80.0 | 6.2 | 6.3 | −0.016 | 7.1 | 7.2 | −0.014 | 8.3 | 8.7 | −0.048 |
| 100.0 | 6.3 | 6.1 | +0.031 | 7.2 | 7.2 | 0 | 8.5 | 8.3 | +0.023 |
| 250.0 | 6.5 | 6.5 | 0 | 7.5 | 7.5 | 0 | 8.8 | 9.0 | −0.022 |
| 400.0 | 6.8 | 7.0 | −0.029 | 7.7 | 7.5 | +0.025 | 9.2 | 9.2 | 0 |
| 600.0 | 7.2 | 7.1 | +0.013 | 8.2 | 8.0 | +0.024 | 10.0 | 9.8 | +0.020 |
| 1000.0 | 8.0 | — | — | 9.1 | — | — | 11.2 | — | — |
| 1500.0 | 9.3 | — | — | 10.5 | — | — | 12.7 | — | — |
| 2000.0 | 10.7 | — | — | 12.1 | — | — | 14.5 | — | — |
| 2500.0 | 12.5 | — | — | 14.3 | — | — | 16.6 | — | — |
| 3000.0 | 14.8 | — | — | 17 | — | — | 19.5 | — | — |

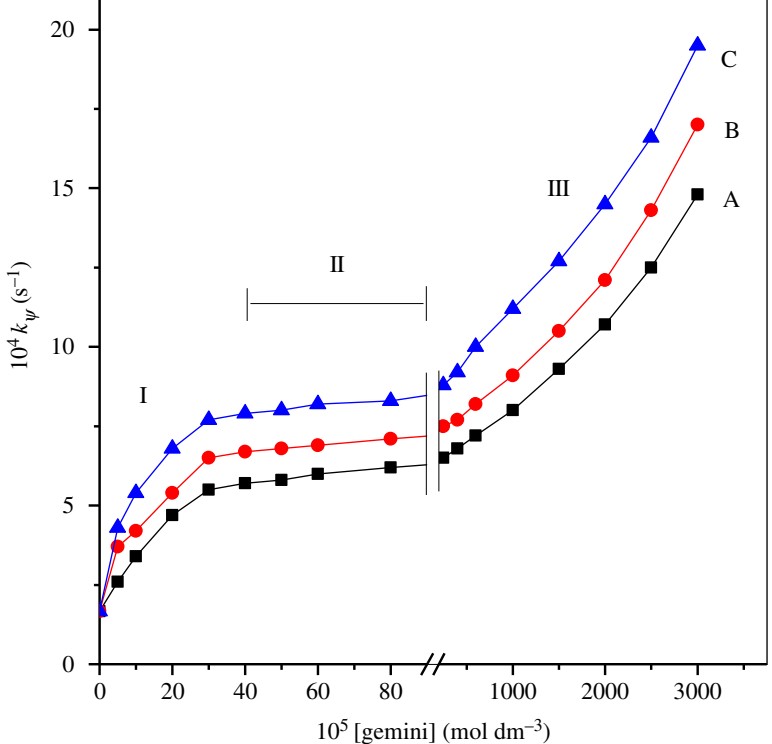

**Figure 3.** Rate constant ($k_\psi$) versus [gemini] on the study of histidine and ninhydrin in surfactants: (A) 16-6-16, (B) 16-5-16 and (C) 16-4-16. Reaction conditions: [His] = $1 \times 10^{-4}$ mol dm$^{-3}$, [ninhydrin] = $6.0 \times 10^{-3}$ mol dm$^{-3}$, temp. = 343 K and pH = 5.0.

**Scheme 2.** Histidine–ninhydrin reaction in aqueous and gemini systems. w, water; m, micelles.

monomeric surfactants. The shape behaviour of Segments I and II are identical to traditional surfactants (single hydrophilic head group and single hydrocarbon tail) [59–62]. The effect of unchanging behaviour of the rate constant in Segment II may be understood when both the substrates are entirely micellar bounded with micellar assembly considered to be staying unaffected [63–65].

In Segment III, the values of the rate constant, $k_\psi$, increase slowly with [surfactant]. Later, a larger increase in $k_\psi$ was observed at higher concentration. This changing in $k_\psi$ is probably owing to variations in the structure of micelles, which is consistent with $^1$H NMR studies of 16-s-16 [37]. Thus, the rate constant increases at higher [16-s-16] caused by variations in micellar morphologies that gives diverse reaction micro-environment (less polar).

## 4.3. Activation parameters

Activation parameters determined on the study of ninhydrin and histidine in gemini micellar system at fixed [reactants] (ninhydrin and histidine), and pH are listed in table 3. Results in table 3 reveal that the presence of gemini catalyses the study more and lowers the values of the enthalpy of activation, $\Delta H^\#$, with a substantial negative entropy, $\Delta S^\#$, in comparison with the corresponding aqueous system. The reduction in $\Delta H^\#$-value follows not only because of the stabilization of the transition state through the

**Table 3.** Activation parameters ($E_a$, $\Delta H^{\#}$ and $\Delta S^{\#}$), rate constant ($k_m$) and binding constants ($K_C$ and $K_D$) determined on the study of ninhydrin ($6 \times 10^{-3}$ M) and histidine ($1 \times 10^{-4}$ M) in aqueous and gemini micellar systems. Uncertainties in thermodynamic parameters $E_a$, $\Delta H^{\#}$ and $\Delta S^{\#}$ are less than or equal to $\pm 0.1$, $\pm 0.1$ and $\pm 0.1$ J K$^{-1}$ mol$^{-1}$, respectively.

| | aqueous | 16-6-16[a] | 16-5-16[a] | 16-4-16[a] |
|---|---|---|---|---|
| $E_a$ (kJ mol$^{-1}$) | 75.3 | 38.1 | 35.8 | 34.3 |
| $\Delta H^{\#}$ (kJ mol$^{-1}$) | 72.5 | 35.3 | 33.0 | 31.5 |
| $-\Delta S^{\#}$ (J K$^{-1}$ mol$^{-1}$) | 308.9 | 312.1 | 313.0 | 314.1 |
| $10^2\, k_m$ (s$^{-1}$)[b] | — | 3.5 | 3.7 | 4.0 |
| $K_C$ (mol$^{-1}$ dm$^3$)[b] | — | 90.0 | 86.0 | 83.0 |
| $K_D$ (mol$^{-1}$ dm$^3$)[b] | — | 58.0 | 56.0 | 53.0 |

[a][16-s-16] = $30 \times 10^{-5}$ mol dm$^{-3}$.
[b]At 343 K.

growth of rigid and well-cultured intermediate molecule but also on account of adsorption of substrate molecule onto the micellar surface. The lowering in entropy leads to the conclusion that the well-cultured activated complex is formed in 16-s-16 than aqueous medium.

# 5. Conclusion

The present study investigates the catalytic influence of gemini surfactants on the rate constant of histidine and ninhydrin at 343 K and pH 5.0 using Shimadzu model UV–vis spectrophotometer. The study of the effect of different constituents on the title reaction including [reactants], temperature and pH was also examined and elaborated in detail. We observe that the presence of geminis even though below their cmc catalyses reaction efficiently when compared with the aqueous system. Gemini surfactants allow them to be employed in lower concentration (cost-effectiveness) to overcome the required catalytic challenges of surfactants in several chemical and industrial applications. They are much better microbiocides when compared with their monomeric analogues because of their wide spectrum of biocidal activity, the safety of applications and low costs [66]. They are used to control and kill harmful and unwanted organisms such as bacteria, mould, algae, insects, etc. Because, gemini surfactants are more effective in disrupting the membrane than the monomeric counterpart, it means that a lesser quantity of gemini, below threshold toxicity, may be required to get the same effect of a considerably higher amount of CTAB. Due to above characteristics, the studied gemini surfactants can be considered as green surfactants. Consequently, they provide less impact on environment. Quantitative treatment of results seems acceptable as the observed rate constant ($k_{\psi}$) and the calculated rate constant ($k_{\psi\text{cal}}$) are in close agreement (table 2).

Data accessibility. Our data are provided as electronic supplementary material.
Authors' contributions. D.K. has done the experiments and written the manuscript. M.A.R. analysed and interpreted data. All authors gave final approval for publication.
Competing interests. The authors declare no competing interest.
Funding. This project was funded by the Deanship of Scientific Research (DSR), King Abdulaziz University, Jeddah, under grant No. D-360-130-1441. The authors, therefore, gratefully acknowledge DSR technical and financial support.

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
