## [Reviewer comments · Royal Society Open Science]

Review History

RSOS-191648.R0 (Original submission)

Review form: Reviewer 1

Is the manuscript scientifically sound in its present form?

Yes

Are the interpretations and conclusions justified by the results?

Yes

Is the language acceptable?

Yes

Do you have any ethical concerns with this paper?

No

Have you any concerns about statistical analyses in this paper?

No

Recommendation?

Accept with minor revision (please list in comments)

Comments to the Author(s)

The paper presents new interesting and useful data and can be published.
Some comments are given below.

- (1) Principal question: is the PIE model applicable when the concentration of the substrate is higher than that of the surfactant?
- (2) The title of section 4.1 looks somewhat unusual.
- (3) Please specify the meaning of the symbols () and [] used in the paper.
- (4) Please indicate the uncertainties (errors, standard deviations) of the numerical results in a more detailed manner.
- (5) 5. Conclusions: can the gemini surfactants be considered as green ones only due to their lower concentrations used in micellar catalysis?

Review form: Reviewer 2

Is the manuscript scientifically sound in its present form?

Yes

Are the interpretations and conclusions justified by the results?

Yes

Is the language acceptable?

Yes

Do you have any ethical concerns with this paper?

No

Have you any concerns about statistical analyses in this paper?

No

Recommendation?

Accept with minor revision (please list in comments)

Comments to the Author(s)

Grammatical and spellings to be checked once again

Decision letter (RSOS-191648.R0)

17-Dec-2019

Dear Dr Kumar:

Title: Catalytic influence of 16-s-16 gemini surfactants on rate constant of histidine and ninhydrin
Manuscript ID: RSOS-191648

Thank you for submitting the above manuscript to Royal Society Open Science. On behalf of the Editors and the Royal Society of Chemistry, I am pleased to inform you that your manuscript will be accepted for publication in Royal Society Open Science subject to minor revision in accordance

with the referee suggestions. Please find the reviewers' comments at the end of this email. I apologise that this has taken longer than usual.

The reviewers and handling editors have recommended publication, but also suggest some minor revisions to your manuscript. Therefore, I invite you to respond to the comments and revise your manuscript.

Please also include the following statements alongside the other end statements. As we cannot publish your manuscript without these end statements included, if you feel that a given heading is not relevant to your paper, please nevertheless include the heading and explicitly state that it is not relevant to your work. We have included a screenshot example of the end statements for reference.

- Funding statement

Please include a funding section after your main text which lists the source of funding for each author.

Because the schedule for publication is very tight, it is a condition of publication that you submit the revised version of your manuscript before 26-Dec-2019. Please note that the revision deadline will expire at 00.00am on this date. If you do not think you will be able to meet this date please let me know immediately.

Supplementary files will be published alongside the paper on the journal website and posted on the online figshare repository (<https://figshare.com>). The heading and legend provided for each supplementary file during the submission process will be used to create the figshare page, so please ensure these are accurate and informative so that your files can be found in searches. Files

on figshare will be made available approximately one week before the accompanying article so that the supplementary material can be attributed a unique DOI.

Best wishes,
Dr Laura Smith
Publishing Editor, Journals

RSC Associate Editor:
Comments to the Author:
(There are no comments.)

RSC Subject Editor:
Comments to the Author:
(There are no comments.)

Reviewer comments to Author:
Reviewer: 1

Comments to the Author(s)

The paper presents new interesting and useful data and can be published.

Some comments are given below.

(1) Principal question: is the PIE model applicable when the concentration of the substrate is higher than that of the surfactant?

(2) The title of section 4.1 looks somewhat unusual.

(3) Please specify the meaning of the symbols () and [] used in the paper.

(4) Please indicate the uncertainties (errors, standard deviations) of the numerical results in a more detailed manner.

(5) 5. Conclusions: can the gemini surfactants be considered as green ones only due to their lower concentrations used in micellar catalysis?

Reviewer: 2

Comments to the Author(s)

Grammatical and spellings to be checked once again

Author's Response to Decision Letter for (RSOS-191648.R0)

See Appendix A.

RSOS-191648.R1 (Revision)

Review form: Reviewer 1

Is the manuscript scientifically sound in its present form?

Yes

Are the interpretations and conclusions justified by the results?

Yes

Is the language acceptable?

Yes

Do you have any ethical concerns with this paper?

No

Have you any concerns about statistical analyses in this paper?

No

Recommendation?

Accept as is

Comments to the Author(s)

The paper is publishable.

But I cannot understand the first phrase of the authors' response. The surfactant concentration in your work was higher as compared with that of the substrate.

Maybe, some misunderstanding took place (?!).

Decision letter (RSOS-191648.R1)

17-Jan-2020

Dear Dr Kumar:

Title: Catalytic influence of 16-s-16 gemini surfactants on rate constant of histidine and ninhydrin
Manuscript ID: RSOS-191648.R1

It is a pleasure to accept your manuscript in its current form for publication in Royal Society Open Science. The chemistry content of Royal Society Open Science is published in collaboration with the Royal Society of Chemistry.

RSC Associate Editor:
Comments to the Author:
(There are no comments.)

RSC Subject Editor:
Comments to the Author:
(There are no comments.)

Reviewer(s)' Comments to Author:
Reviewer: 1

Comments to the Author(s)

The paper is publishable.

But I cannot understand the first phrase of the authors' response. The surfactant concentration in your work was higher as compared with that of the substrate. Maybe, some misunderstanding took place (?!).

Appendix A

Journal Title: Royal Society Open Science

Manuscript Title: Catalytic influence of 16-s-16 gemini surfactants on rate constant of histidine and ninhydrin

Manuscript ID: RSOS-191648

Dear Professor Smith,

Thank you very much for your useful comments. We have modified the manuscript in the light of Reviewers' comments. Detailed corrections are listed below point by point.

Thanking you,

Sincerely yours,

Dr. Dileep Kumar

Response to Reviewer # 1:

1. Thank you very much for your comment. PIE model is not applicable when the concentration of the substrate is higher than that of the surfactant.

The reaction occurs between histidine and ninhydrin under pseudo-first-order condition using \geq 10-fold excess of [ninhydrin] over [Histidine]. Study carried out in two media (aqueous and surfactant). Effect of 16-s-16 surfactant was performed on the reaction and rate constants were obtained by varying 16-s-16 surfactant concentrations ranging from $(5-3000) \times 10^{-5} \text{ mol dm}^{-3}$ keeping other reaction factors constant. We observed that presence of geminis even though below their cmc catalyzed reaction efficiently as compared to aqueous system. Present study in presence of gemini surfactants can only be explained by the means of pseudo-phase model of micellar catalysis suggested by Menger and Portnoy [1] and established by Bunton [2] and Romsted [3].

- (1) Menger FM, Portnoy CE. 1967 Chemistry of reactions proceeding inside molecular aggregates. *J. Am. Chem. Soc.* **89**, 4698-4703. (doi:10.1021/ja00994a023). (2) Bunton CA. 1979 Reaction kinetics in aqueous surfactant solutions, *Catal. Rev. Sci. Eng.* **20**, 1–56. (doi:10.1080/03602457908065104). (3) Romsted LS. 1977 *Micellization, solubilization, and microemulsions*. In: Mittal KL, vol. 2. New York: Plenum Press.

2. Now, title of section 4.1 is modified (Pl. see on Page 6).
3. Changes are made as suggested (Pl. see on Page 2, line 12 & 17; Page 3, line 29; Page 4, line 29; Page 5, line 30; Page 6, line 1; Page 7, line 6; Page 8, lines 2,4 & 11; Page 9, lines 2, 9, 23 & 26 and Page 10, line 11).
4. Required information is given (Pl. see Page 18, Table 1; Page 20, Table 2 and Page 21, Table 3).
5. Necessary explanation is provided in conclusion section (Pl. see on Page 10).

Response to Reviewer # 2:

Thank you very much for your comments. Now, grammatical and spelling errors are removed (Pl. see on Page 2, line 21; Page 3, line 7; Page 5, line 5 & 19; Page 8, line 8 and Page 10, line 1).